# A Titanium Tetrafluoride Experimental Gel Combined with Highly Concentrated Hydrogen Peroxide as an Alternative Bleaching Agent: An *In Vitro* Study

**DOI:** 10.3390/gels8030178

**Published:** 2022-03-14

**Authors:** Rodrigo Lins, Pedro Rosalen, Diego Silva, Bianca Kawabata, Luís Roberto Martins, Vanessa Cavalli

**Affiliations:** 1Department of Restorative Dentistry, Operative Dentistry Division, Piracicaba Dental School, University of Campinas, Avenida Limeira 901, Bairro Areião, Piracicaba 13414-903, SP, Brazil; biancaarissa_@hotmail.com (B.K.); martins@unicamp.br (L.R.M.); cavalli@unicamp.br (V.C.); 2Department of Pathology and Parasitology, Federal University of Alfenas, Rua Gabriel Monteiro da Silva, Centro, Alfenas 37130-001, MG, Brazil; rosalen@fop.unicamp.br; 3Department of Biosciences, Piracicaba Dental School, University of Campinas, SP. Avenida Limeira 901, Bairro Areião, Piracicaba 13414-903, SP, Brazil; 4Research Program in Integrated Dental Sciences, University of Cuiabá, Cuiabá 78065-900, MG, Brazil; diegoromarioo@gmail.com

**Keywords:** bleaching agents, tooth bleaching, fluorides, titanium tetrafluoride, fluorides topical

## Abstract

This *in vitro* study evaluated color change, mineral content, and morphology of enamel, pH and cytotoxicity of experimental bleaching agents containing 35% hydrogen peroxide (HP), titanium tetrafluoride (TiF${}_{4}$), Natrosol, and Chemygel. Sixty enamel/dentin blocks were randomly treated with (n = 10) HP; HP+Natrosol+Chemygel with different TiF${}_{4}$ concentrations: 0.05 g HPT0.5, 0.1 g HPT1, 0.2 g HPT2, 0.3 g HPT3, 0.4 g HPT4. Bleaching was performed in three sessions (3 × 15 min application). Color change (CIEL${}_{ab}$-Δ$E_{ab}$, CIEDE2000-Δ$E_{00}$, Δ$WI_{D}$) and Knoop microhardness (KHN) were evaluated. Enamel morphology and composition were observed under scanning electron microscopy and energy-dispersive spectrometry (EDS), respectively. Cell viability of keratinocyte cells was evaluated using MTT assay. Data were analyzed by one-way ANOVA and LSD and Tukey tests, and two-way repeated measures ANOVA and Bonferroni (α = 5%). The pH and EDS were analyzed descriptively. Lightness-L* increased, and a* and b* parameters decreased, except for HPT3 and HPT4 (b*). HPT0.5, HPT1, and HPT2 exhibited Δ$E_{ab}$ and Δ$WI_{D}$ similar to HP. Δ$E_{00}$ did not present statistical difference. HP, HPT0.5, and HPT1 promoted higher KHN. HPT0.5 exhibited no changes on enamel surface. Keratinocyte cells were viable when treated with T0.5, and weak viable for T1. Experimental agents exhibited acidic pH and Ti elements. HPT0.5 exhibited bleaching efficacy, maintained KHN without enamel alterations, and did not increase cytotoxicity.

## 1. Introduction

The expressive demand for dental bleaching to improve patients’ aesthetic appearance has grown considerably over the past years [1]. The procedure can be performed in a clinical setting with high concentrations of bleaching agents (35–40% hydrogen peroxide, HP, or carbamide peroxide, CP) under professional supervision, or at home, with low concentrations of peroxide agents (3–10% HP or 10–20% CP). Both techniques promote satisfactory clinical bleaching efficacy (10% CP vs. 38% HP) [2,3,4].

Tooth color is determined by hue, value, chroma, thickness, texture, and translucency of enamel and dentin [5]. These factors are modulated by either extrinsic (through contact with staining substances adhering to the tooth) or by intrinsic causes (through structural changes in the dental hard tissues composition or thickness during tooth development) [5,6]. In both cases, bleaching is effective because HP decomposes into free radicals that diffuse from enamel to dentin, and are able to oxidize complex organic molecules into smaller-sized ones, producing the whitening effect due to light reflectance difference [7,8,9]. Although bleaching is clinically effective [2], some adverse events, such as tooth sensitivity and gingival irritation during and after bleaching procedure [2,10], structural and morphological enamel alterations [11], and mineral loss [12], are frequently observed.

In order to overcome the morphological changes or mineral loss caused by bleaching procedures, topical fluoride application after bleaching [13], or the incorporation of high-concentrations of NaF into bleaching agents, have been described, aiming to reduce the risk of tooth sensitivity [14] or to promote enamel remineralization by mineral compounds [15,16].

Our research group recently developed an agent containing titanium tetrafluoride (TiF${}_{4}$), Natrosol, and Chemygel to be incorporated into 35% HP bleaching gel [17]. This agent was designed for whitening purposes and to maintain enamel integrity through the bleaching procedure. TiF${}_{4}$ is frequently used in solutions or incorporated into varnishes, and according to previous research it deposits on enamel surface aiming to reverse or control enamel demineralization promoted by caries [18,19] or erosion lesions [20]. It has been speculated that TiF${}_{4}$ minimizes demineralization as it tends to complex with the phosphate of the enamel apatite, forming a rich-titanium oxide or hydrated titanium phosphate glazed layer [21,22].

In our previous study [17], an experimental gel-based formula (Natrosol with Chemygel) containing 4% TiF${}_{4}$ associated with a commercial 35% HP agent was evaluated and we have found that this combination exhibited, in general, similar results compared with 35% HP alone. The experimental agent increased microhardness at the end of bleaching, and no enamel surface alterations were observed [17]. However, since that study was the first using this combination, we needed to narrow TiF${}_{4}$ concentration range and examine the efficacy of the agents according to enamel color changes, surface mineral content, morphology, and evaluate cytotoxicity of the experimental agent applied on keratynocytes cells (HaCat) [23], since oral mucosa irritation is also a common effect of bleaching procedure [24].

Based on the exposed, this *in vitro* study evaluated enamel color changes, surface microhardness, and morphology after bleaching with experimental agents containing low concentrations of TiF${}_{4}$ and cell viability of HaCat cells treated with the experimental agents. The null hypotheses were that (1) the experimental agents containing TiF${}_{4}$ would promote whitening effect, regardless of TiF${}_{4}$ concentration; (2) the experimental agents containing 0.05 g or 0.1 g of TiF${}_{4}$ concentrations would not modify enamel surface microhardness; (3) experimental agents would not negatively influence enamel surface morphology; and (4) the experimental agents would not promote cytotoxicity to HaCat cells.

## 2. Materials and Methods

### 2.1. Specimen Preparation

Sixty enamel/dentin bovine blocks were obtained from buccal surface of sound bovine incisors crowns. The bovine incisors were cut with a diamond blade (Buehler, Lake Bluff, IL, USA) to obtain blocks with standard dimensions (5 × 5 mm). Enamel surface was polished with silicon carbide papers (600, 1500, and 4000 grit) (Buehler, Lake Bluff, IL, USA), intercalated with ultrasonic baths with distilled water for 10 min (Marconi, Piracicaba, SP, Brazil). Dentine surface was isolated with transparent acid-resistant varnish (L’Apogée Alfaparf, Campo Grande, RJ, Brazil) allowing only enamel surface exposure. Specimens were immersed in black tea solution (Dr. Oetker, São Paulo, SP, Brazil) for 24 h at room temperature (25 ${}^{\circ }$C), followed by water storage for 7 days, with daily exchanges [25]. All polished specimens were submitted to microhardness indentations (Shimadzu, Kyoto, Japan) to determine enamel initial Knoop microhardness number (KHN). Enamel blocks with mean values of 278.66 kg/mm${}^{2}$ ± 21.63 were selected and randomly allocated into six treatment groups (n = 10), as described in Table 1.

### 2.2. Experimental Bleaching Gel Preparation and Treatment Protocol

Our research group recently developed an experimental gel containing TiF${}_{4}$, Natrosol, and Chemygel [17], and this formulation was the base for the experimental groups displayed in Table 1. The experimental gel contains Natrosol and Chemygel in different proportions, correspondent with different concentrations of TiF${}_{4}$ (0.05, 0.1, 0.2, 0.3, and 0.4 g of TiF${}_{4}$). The reagents were weighed with analytical precision balance (Chyo JEX-200, YMC Co Ltda, Tokyo, Japan), manipulated in plastic container with plastic stick, and vortexed until complete homogenization. The control group was represented by a commercial 35% hydrogen peroxide HP agent (Whiteness HP 35% FGM, Joinville, SC, Brazil), and application followed the manufacturer’s instructions. The experimental gel containing TiF${}_{4}$ was incorporated into the commercial bleaching agent (HP) immediately before application. All bleaching gels were applied on enamel surface three times for 15 min. After dental bleaching protocol, the specimens were rinsed with distilled water and stored in artificial saliva (1.5 mM CaCl, 0.9 mM NaH${}_{2}$PO${}_{4}$, 150 mmol/L KCl, pH 7.0) at 37 ${}^{\circ }$C [26], renewed every 24 h. Bleaching protocol was performed in three sessions of 72 h intervals [27,28].

### 2.3. Color Measurement

Color measurements were performed before bleaching (baseline) and 24 h after each bleaching session [27]. The digital manual spectrophotometer Vita Easyshade (Vita-Zahnfabrik, Bad Säckingen, Germany) was used for color measurement in triplicate. The manual spectrophotometer was clamped to a three-fingered laboratory clamp, and the specimen was positioned over an opaque white ceramic background. The specimen and the ceramic background were lifted (Jack lift-Q219, Quimis) to meet the spectrophotometer’s tip. Color evaluation was performed in a controlled light environment with three measurements at different positions of the ceramic background (0, 45, and 90${}^{\circ }$ of rotation). Color variation was determined by CIEL${}_{ab}$ parameters (L*, a*, and b*), color changes (Δ$E_{ab}$) (1) [29], CIEDE2000 color difference (Δ$E_{00}$) (2) [30], and whiteness index difference (Δ$WI_{D}$ = $WI_{D}$ final −$WI_{D}$ baseline) (3 and 4) [31], according to the following equations: (1)$$\Delta E_{ab}={\left[{\left(\Delta L*\right)}^{2}+{\left(\Delta a*\right)}^{2}+{\left(\Delta b*\right)}^{2}\right]}^{1/2}$$
(2)$$\Delta E_{00}={\left[{\left(\Delta L/K_{L}S_{L}\right)}^{2}+{\left(\Delta C/K_{C}S_{C}\right)}^{2}+{\left(\Delta H/K_{H}S_{H}\right)}^{2}+RT*\left(\Delta C/K_{C}S_{C}\right)*\left(\Delta H/K_{H}S_{H}\right)\right]}^{1/2}$$
(3)$$WI_{D}=0.511L*-2.324a*-1.100b*$$
(4)$$\Delta WI_{D}=WI_{D}4\mathrm{th}\;\;application-WI_{D}baseline$$

Legend: $\Delta E_{ab}$, color changes; ΔL, lightness difference; Δa, green–red coordinate difference; Δb, blue–yellow coordinate difference; *L*, Lightness; *a*, green-red coordinate; *b*, blue-yellow coordinate; $\Delta E_{00}$, CIEDE2000; ΔC, chroma coordinate difference; ΔH, hue difference; $S_{L}$, $S_{C}$, and $S_{H}$, weighting functions; $K_{L}$, $K_{C}$, and $K_{H}$, parametric factors; RT, rotation function; $WI_{D}$, whiteness index; Δ$WI_{D}$, whiteness index differences.

### 2.4. Microhardness Analysis

Knoop microhardness (Shimadzu, Kyoto, Japan) was performed four times, at baseline for specimen selection and 24 h after each bleaching application. Five indentations were made 500 μm away from enamel margins and 100 μm apart under 490.3 mN load for 10 s. Mean values of the five measurements were obtained [18].

### 2.5. pH Measurement

Commercial and experimental bleaching agents were submitted to pH measurement (Thermo Electron Corporation, Waltham, MA, USA). A pH meter was calibrated with standard solutions (pH 1.0, 4.0, and 7.0), and measurement was performed for each bleaching agent in triplicate, at different times (0, 5, 10, and 15 min), using a small electrode device immersed in gel preparations [32].

### 2.6. Scanning Electron Microscopy (SEM) and Energy-Dispersive Spectrometry (EDS)

Three specimens of each group were selected and observed under SEM (JEOL-JSM, 6460LV, Tokyo, Japan). The specimens were washed in ultrasonic bath (Ultra Cleaner, Unique, Indaiatuba, SP, Brazil) for 10 min and dried for 24 h. After drying, specimens were sputter-coated with gold (MED 010, Balzers, Balzer Liechtenstein) and submitted to SEM evaluation (3000× magnification) in vacuum mode (45 Pa), operating at 15 kV [18]. For EDS analysis, the other three specimens were covered with carbon and submitted to automatic image analyzer system (JEOL-JSM, 6460LV, Tokyo, Japan), providing a percentage of the chemical elements (%atomic) present on the total area of the specimens’ surfaces [18].

### 2.7. Toxicity on HaCaT Cells In Vitro

In order to provide evidence on the potential toxic effects of the formulations, the MTT (3-(4,5-dimethylthiazol-2-yl)-2,5-diphenyltetrazolium bromide) viability test with keratinocytes (HaCat) was conducted [33]. The HaCat cells were grown in DMEM (Dulbecco’s Modified Eagle Medium, Thermo Fisher Scientific, Waltham, MA, USA) supplemented with 10% fetal bovine serum (Sigma-Aldrich, San Luis, MO, USA). Cells were seeded onto 96-well plates at a density of 2 × 105 cells/well and allowed to adhere for 24 h. Then, an extract of the samples was obtained through the contact between the gel and DMEM, as previously described for another solid dental material [34]. Concentrations ranged from 0.05 to 0.4 g TiF${}_{4}$. Posteriorly, the adhered cells were exposed for 15 min to the extract of the experimental formulations. A vehicle-treated group was also included in this assay for comparison. After incubation, viability was determined by adding 200 μL of MTT solution (0.3 mg/mL) into the wells. The precipitated formazan crystals were solubilized in ethanol, and the absorbance was set at 570 nm using a microplate reader [35]. Cell viability was classified by ISO 10993-5 into four scores of citotoxicity: non-citotoxicity (>80%), weak (60–80%), moderate (40–60%), and strong (<40%) [36].

### 2.8. Statistical Analysis

Data were tested for normal distribution and homoscedasticity (Shapiro–Wilk/Levene). CIEL${}_{ab}$ parameters (L*, a*, b*) and microhardness were analyzed by two-way repeated measures, ANOVA and Bonferroni post hoc, for experimental group comparison. Whiteness index differences (Δ$WI_{D}$), color alteration (Δ$E_{ab}$), and CIEDE2000 color difference (Δ$E_{00}$) were tested by one-way ANOVA and LSD post hoc test. Cell viability was tested by one-way ANOVA and Tukey post hoc test. Statistical analyses were performed by SPSS 21.0 (SPSS Inc., Chicago, IL, USA), with significance level of 5%. Then, pH and EDS measurements were submitted to descriptive analyses.

## 3. Results

### 3.1. Color Measurement

Lightness (L* coordinate) increased after bleaching protocol for all treatments (*p* < 0.035). However, at the end of bleaching (third session), no differences in lightness were observed among the treatment groups (*p* > 0.05). Mean values of a* coordinate decreased after the first bleaching application for all groups (*p* < 0.032), indicating decrease of the red color. At the end of bleaching, HP exhibited no difference in a* values compared to HP, HPT0.5, HPT1, and HPT2 (*p* > 0.05), but higher a* values than HPT3 and HPT4 (*p* < 0.012). The b* coordinate analysis indicated that the enamel yellow appearance of HP, HPT0.5, HPT1, and HPT2 decreased throughout bleaching (*p* < 0.016) compared with baseline. After the third bleaching application, HP promoted lower b* values than HPT2, HPT3, and HPT4 (*p* < 0.04), but no differences were observed for HPT0.5 and HPT0.1 (*p* > 0.05) (Figure 1). HP exhibited no difference in Δ$E_{ab}$ compared to HPT0.5, HPT1, and HPT4 (*p* > 0.05), but higher than HPT2 (*p* < 0.025) and HPT3 (*p* < 0.032). No differences in Δ$E_{00}$ were observed among groups (*p* > 0.05). HP exhibited no differences in Δ$WI_{D}$ compared to HPT0.5, HPT1, and HPT2 (*p* > 0.05), but higher than HPT3 and HPT4 (*p* < 0.036) (Figure 2).

### 3.2. Surface Microhardness

At baseline, no differences in microhardness values among groups were found (*p* > 0.05) (Figure 3). HPT2, HPT3, and HPT4 decreased microhardness values from baseline to the third bleaching application (*p* < 0.001). At the end of bleaching therapy, no differences were found among HP, HPT0.5, and HPT1 (*p* > 0.05), and these groups exhibited higher microhardness values than HPT2, HPT3, and HPT4 (*p* < 0.047).

### 3.3. pH Measurement

Table 2 exhibits the results of pH measurements of each experimental agent combined with HP at different times (0, 5, 10, and 15 min of bleaching). HP group presented higher pH values than the experimental agents at all evaluation times (6.66, 6.01, 5.83, and 5.62, respectively), and HPT0.5 was the agent with pH values closer to HP (5.43, 4.89, 4.77, and 4.82, respectively). HPT4 exhibited lower mean pH values among groups at all times. All the experimental agents exhibited acidic pH values, including the control group, and the higher the TiF${}_{4}$ concentrations, the more acidic the experimental agent.

### 3.4. SEM and EDS

Enamel surface morphology was analyzed by SEM and EDS and is displayed in Figure 4. The HP control group exhibited a discontinued surface with mild alterations, irregular areas, and absence of Ti element. For experimental HPT0.5 and HPT1 groups, surface was smoother and more regular, with no signs of demineralization. However, as TiF${}_{4}$ concentration increased, surfaces exhibited areas with enamel prism exposure (more pronounced after HPT3 and mainly after HPT4 treatments). The increasing concentration of Ti elements was directly related to the amount of TiF${}_{4}$ incorporated into the bleaching agent. Ca and P were also detected by EDS and exhibited no significant percentage alterations.

### 3.5. Toxicity on HaCaT Cells In Vitro

Cytotoxicity of HaCAt cells treated with TiF${}_{4}$ with Natrosol and Chemygel (without HP) is represented in Figure 5. HaCat treated with HPT0.5 was considered non-citotoxic (>80%), with significant difference between HPT1 (weak cytotoxicity, 60–80%) (*p* < 0.001) and other groups (*p* < 0.001). The higher TiF${}_{4}$ concentrations (0.2 g, 0.3 g, and 0.4 g) severely decreased cell viability (<40%). Figure 6 represents HaCat viability of experimental agents containing HP and the control group. All bleaching agents containing 35% HP significantly decreased cell viability and were toxic to HaCat cells (<40%) (*p* > 0.05).

## 4. Discussion

This investigation analyzed dental bleaching ability and mineral content of enamel submitted to experimental 35% HP bleaching agents containing low concentrations of titanium tetrafluoride (0.05–0.4 g). In a previous investigation [17], our research group observed that 4% TiF${}_{4}$ (or 0.198 g) combined with Natrosol and Chemygel thickeners exhibited results similar to the HP commercial bleaching agent. This study selected the best TiF${}_{4}$/thickener combination outcome of the first research (0.198 g TiF${}_{4}$), and concentrations of TiF${}_{4}$ ranged around the known concentration (0.05, 0.1, 0.2, 0.3, and 0.4 g).

Bleaching treatment with control HP, or experimental agents, increased enamel lightness (L*) and decreased a* values. Since enamel was stained with black tea solution, the decrease of a* values (from red to green) could indicate the ability of agents to remove extrinsic tea stains. However, at the end of treatment, HPT3 and HPT4 exhibited greater capacity of decreasing a* values than HP, which was similar to HPT0.5, HPT1, and HPT2.

The ability to decrease enamel yellow appearance (b* values) was noted for HP-, HPT0.5-, HPT1-, and HPT2-treated groups. In the opposite direction, HPT3 increased b* values and HPT4 remained similar to baseline. Furthermore, HPT3 caused lower color change (Δ$E_{ab}$) and lower whiteness index values (Δ$WI_{D}$) than HP group. Although enamel color change of HPT3- and HPT4-treated groups were higher than the clinically acceptable value of 2.7 [37], instead of whitening, these agents noticeably turned enamel surface yellow. Because of this outcome, these concentrations (HPT3 and HPT4) were considered inappropriate for whitening purposes. Therefore, the first hypothesis was rejected, because experimental dental bleaching with higher TiF${}_{4}$ concentrations (HPT3 and HPT4) did not promote dental whitening.

In general, the experimental agents that promoted enamel whitening effect (HPT0.5, HPT1, and HPT2) exhibited color changes similar to HP, such as lightness increase (L*), a* and b* values decrease, similar whiteness index (Δ$WI_{D}$), and color changes (Δ$E_{ab}$) along bleaching applications. Furthermore, these agents apparently did not interfere in the decomposition of HP into free radicals (hydrogen, hydroxyl, and perhydroxyl radicals) [38].

Although experimental agents should present a whitening effect in order to be suitable for use, the goal of incorporating TIF${}_{4}$ was to keep enamel mineral content through bleaching. HPT2, HPT3, and HPT4 agents significantly decreased enamel microhardness, and at the end of treatment, exhibited lower mineral content than the other groups. Although HPT2 promoted color change comparable with control group, this agent was also incapable of controlling enamel mineral content after bleaching. Therefore, HPT2 could not be indicated for treatment either. On the other hand, HPT0.5 and HPT1 agents kept enamel mineral content through bleaching, as microhardness values were similar to HP. Therefore, the second null hypothesis could be accepted as that low TiF${}_{4}$ concentrations (0.05 g and 0.1 g) did not decrease microhardness compared with HP.

The behavior of the experimental agents on enamel microhardness is directly correlated with different TiF${}_{4}$ concentrations. One study evaluated TiF${}_{4}$ solutions at different concentrations (1–4%) applied on dentin with carious lesions and reported that the lower concentrations were more effective in remineralizing dentin surface [19].

It is believed that the ability of TiF${}_{4}$ to remineralize enamel relies on the formation of an acid-resistant Ti layer on the surface that can be more effective than NaF [39]. Additionally, according to previous observations, it was noted that this vitreous layer is not homogenously formed due to regional differences of Ca and P content in different enamel areas [18]. Our EDS results indicated that the presence of Ti on enamel treated with the experimental agents and Ti concentration was higher with higher TiF${}_{4}$ content. However, additional analysis should be performed in order to identify and map Ti layer distribution over enamel.

SEM images combined with EDS mineral content analysis showed, as expected, that the higher the TiF${}_{4}$ concentration, the greater the enamel morphological changes. HPT4 promoted severe alterations resembling enamel demineralization. HP and HPT2 also presented enamel morphology changes, however not as harsh as HPT3 and HPT4. The intermediate morphological alterations exhibited by HP could be related to mineral loss content during bleaching protocol (within bleaching sessions) and the remineralizing effect promoted by artificial saliva, since microhardness of this group did not decrease throughout bleaching [40]. On the other hand, surface alterations promoted by HPT4 were possibly so severe that artificial saliva was not able to reverse it. Contrary to these observations, lower TiF${}_{4}$ concentrations (HPT0.5 and HPT1) kept enamel morphologically intact. Due to these results, the third hypothesis was rejected, as higher TiF${}_{4}$ concentrations changed enamel morphology.

The experimental gels used (0.05, 0.1, 0.2, 0.3, and 0.4 g TiF${}_{4}$ with Natrosol and Chemygel) presented low pH values after 15 min of bleaching (4.82, 4.29, 4.20, 3.19, and 2.10, respectively), whereas HP exhibited pH of 5.61. It is known that pH values below the critical value established for enamel (<5.5) trigger enamel mineral loss [41]. However, it was also observed that enamel mineral dissolution increases porosity, allowing deep deposition of calcium, phosphate, and fluoride ions [19]. This probably occurred with the lower concentration (HPT0.5), however, was not observed with the higher TiF${}_{4}$ content agents. In fact, the highly acidic pH of HPT3 and HPT${}_{4}$ possibly promoted enamel mineral loss and morphological alterations, as observed by one study [19].

In this study, the pathological effects of HP and the experimental agents were analyzed on HaCat cells as a model [33]. Although HP and the experimental agents are displayed in gel, which effectively reduces risk of high dose exposure to oral mucosa compared to the liquid form [33,42], the long-term exposure to high doses of HP can damage soft and hard oral tissue [43]. However, the effects of the experimental agents combined with HP were unknown.

Cell viability was analyzed by MTT assay, and the application of 0.05 g of TiF${}_{4}$ with Natrosol and Chemygel (without HP, Figure 5) was not cytotoxic, and HaCat cells were >80% viable [36]. After HaCat had been treated with extract of 0.1 g of TiF${}_{4}$ with Natrosol and Chemygel, weak viability was expressed (60–80%), but this outcome was still positive as the critical percentage value of toxicity is below 40% [36]. On the other hand, cells treated with 0.2, 0.3, and 0.4 g of TiF${}_{4}$ with Natrosol and Chemygel were significantly less viable, and these agents promoted severe cytotoxic effects, decreasing the metabolic activity (<40%). Nonetheless, the incorporation of HP into the experimental agents on HaCat cells promoted severe cell toxicity even for the HP-treated median, as cells treated with HP, HPT0.5, HPT1, HPT2, HPT3, and HPT4 were significantly less viable (<40%) (Figure 6). These results are in accordance with the recent findings of one study [33], in which the authors exposed primary cultured normal human oral keratinocytes (NHOKs) to different doses and application times of HP. The authors used doses ranging from 0.01–100 mM of HP, which is about 1/10 of that of 3% HP (home-applied agent). According to their results, significant cellular damage was observed when dose exceeded 5 mM, and severe cytotoxicity was observed with 15 min of exposure with treatment doses higher than 100 mM [33]. In our research, the extract used in the media (following another study) [34] corresponds to the exact amount that would be applied on enamel in an in-office application and could contact keratinocytes. Therefore, since HP is known for its highly cytotoxic effects, even in very low-concentrations [33], the absence of cell viability after the addition of HP to the experimental agents was expected due to the highly oxidative capacity of H${}_{2}$O${}_{2}$ and the high concentrations used (35% H${}_{2}$O${}_{2}$ instead of 3% H${}_{2}$O${}_{2}$).

It is scientifically established that high concentrations of HP damage cells, and it is directly related to the bleaching exposure time [44]. It should be noted, however, that cytotoxicity was promoted by 35% HP addition, but the experimental agent with 0.05 g TiF${}_{4}$ with Natrosol and Chemygel maintain cell viability through treatment. Based on this result, we rejected the fourth hypothesis, since only the experimental agent containing 0.05 g and 0.1 g of TiF${}_{4}$ maintain cell viability when not combined with HP. This preliminary report concerning HaCat cell viability must be further complemented in order to properly understand cellular damage involving other cell types, and other methodologies [27]. However, it should also be pointed out that 35% HP is applied over enamel surface and a light-cured resin barrier is applied over gingival margins in order to protect it. Since the experimental agent 0.05 g TiF${}_{4}$ with Natrosol and Chemygel was not toxic to HaCat cells, it seems that no differences in technical procedures would be made in case this agent is clinically used in combination with HP.

The limitation of this study was related to cell viability methodology performed in HaCat cells, which represents an evaluation of accidental contact of dental bleaching gel during in-office application technique. Evaluation in odontoblast cells needs to be performed in future studies to determine the protective effect of this experimental titanium tetrafluoride compound-based gel to determine safety and prevent tooth sensibility.

The first studies concerning enamel alterations promoted by high-concentrated bleaching agents frequently exhibited a porous surface [45] with roughness increase [46], and signs of mineral content loss [47,48]. On the other hand, formulations of the latest bleaching products are different from those of the original versions. In fact, manufacturers have improved composition of agents by changing the thickeners used and pH of the gel [49,50,51]. These modifications in HP formula have minimized surface morphology changes and/or controlled enamel mineral content. However, as our results show, they still exist, as HP promoted minor morphological changes.

Although enamel changes promoted by HP could be clinically reversed by saliva [52], concerns exist regarding the kind of substrate that bleaching would be performed on: sound, erosion like-surface [53], white spot lesions [54], and minor enamel cracks or defects [55]. Therefore, the experimental blend containing 0.05 g TiF${}_{4}$ with Natrosol and Chemygel combined with 35% HP could control surface microhardness and enamel morphology, displaying low cytotoxicity, without compromising the ability of HP to bleach.

## 5. Conclusions

The combination of a commercial dental bleaching agent (35% HP-based gel) and a experimental gel containing 0.05 g or 0.1 g of TiF${}_{4}$, Natrosol, and Chemygel could be an alternative to conventional in-office bleaching, as these agents were able to control surface mineral content without compromising enamel morphological or whitening. In addition, the experimental gel containing 0.05 g or 0.1 g TiF${}_{4}$ with Natrosol and Chemygel (without HP) did not exhibit low cytotoxicity on keratinocytes cells, respectively.

## Figures and Tables

**Figure 1 gels-08-00178-f001:**
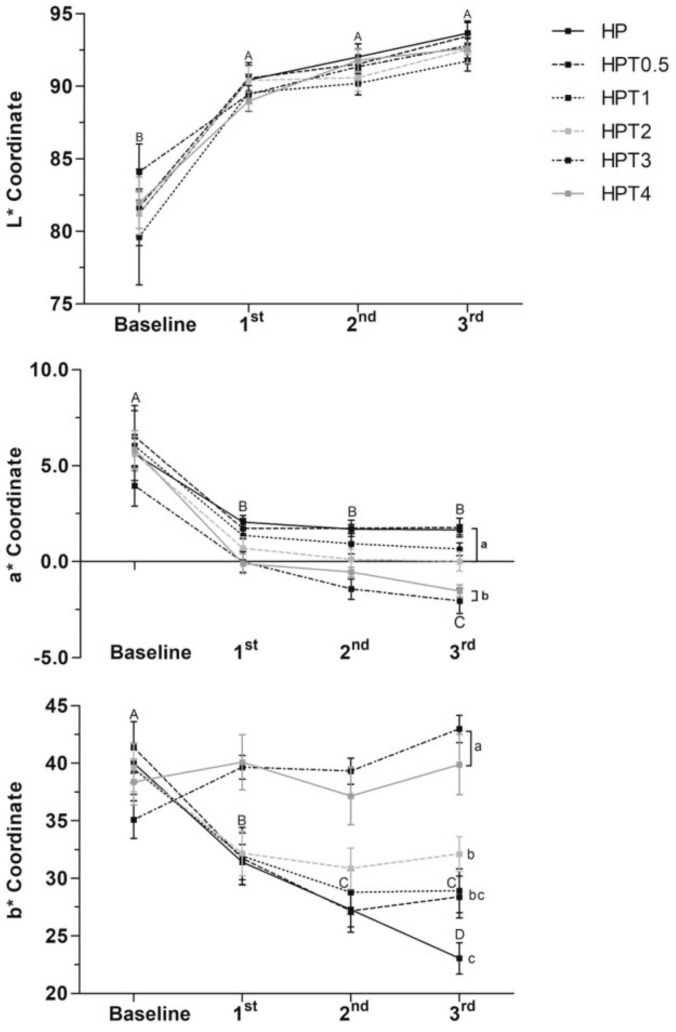
CIEL*a*b* coordinates of different bleaching treatment over time application. Legend: Means and standard deviation followed by different letters indicate statistical differences according to two-way repeated measures, ANOVA and Bonferroni test, with significant level set at 5%. Uppercase letters compare color parameters over time within bleaching agents. Lowercase letters compare bleaching treatment groups. N = 10 specimens/group. HP: hydrogen peroxide; T: titanium tetrafluoride.

**Figure 2 gels-08-00178-f002:**
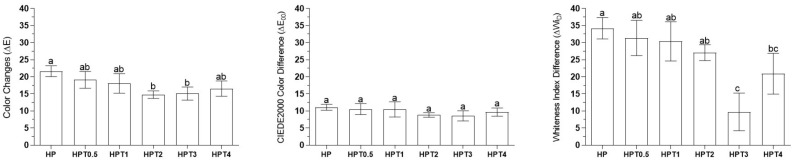
Color change (Δ$E_{ab}$), CIEDE2000 color difference (Δ$E_{00}$), and whiteness index difference (Δ$WI_{D}$) after bleaching treatment. Legend: Means and standard deviation followed by different letters indicate statistical differences according to one-way ANOVA and LSD post hoc test with significant level set at 5%. Lowercase letters compare bleaching treatment groups. N = 10 specimens/group. HP: hydrogen peroxide; T: titanium tetrafluoride.

**Figure 3 gels-08-00178-f003:**
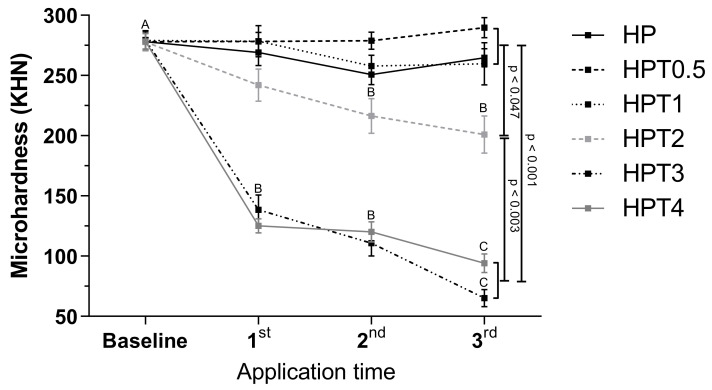
Surface microhardness (KHN) of different bleaching treatment 24 h after application. Legend: Means and standard deviation followed by different letters indicate statistical differences according to two-way repeated measures, ANOVA and Bonferroni test, with significant level set at 5%. Uppercase letters compare color parameters over time within bleaching agents. *p*-Values indicate statistically significant difference of bleaching treatment groups at third application time. N = 10 specimens/group. HP: hydrogen peroxide; T: titanium tetrafluoride.

**Figure 4 gels-08-00178-f004:**
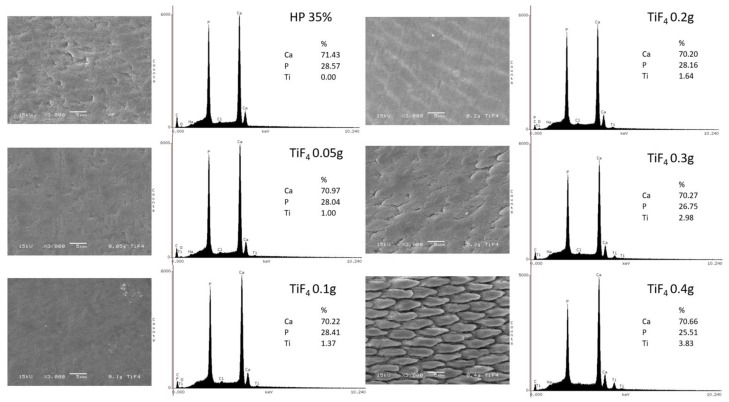
Enamel surface morphology after bleaching treatment of each group by SEM (3000× magnification) and EDS analysis. Legend: Ca: calcium; P: phosphorus; Ti: titanium; TiF${}_{4}$: titanium tetrafluoride.

**Figure 5 gels-08-00178-f005:**
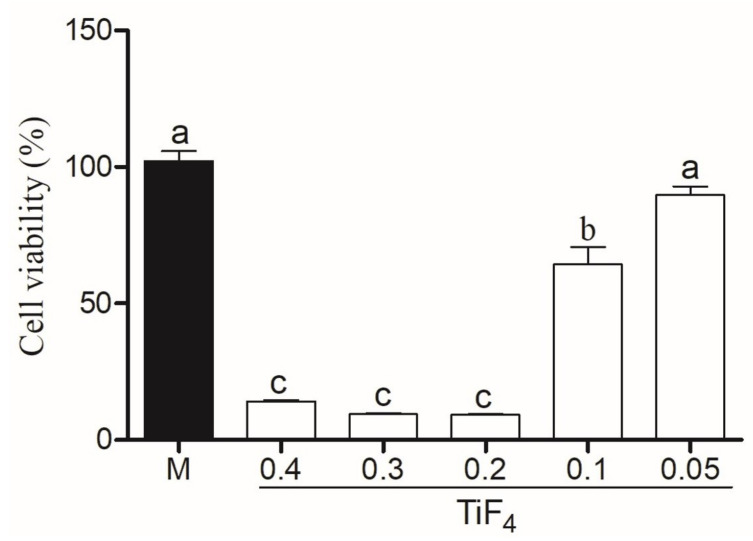
Cell viability of experimental gel with Natrosol, Chemygel, and different concentrations of TiF${}_{4}$ (0.4, 0.3, 0.2, 0.1, and 0.05 g) compared to medium (control group). Legend: Means followed by different letters indicate statistical differences according to one-way ANOVA with Tukey post hoc test with significant level set at 5%. M: medium; HP: hydrogen peroxide; TiF${}_{4}$: titanium tetrafluoride.

**Figure 6 gels-08-00178-f006:**
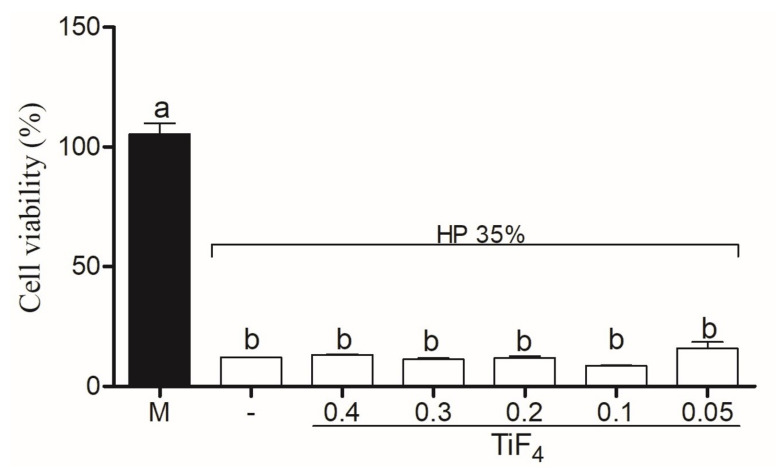
Cell viability of control group (HP) and treatment groups combined with HP. Legend: Means followed by different letters indicate statistical differences according to one-way ANOVA with Tukey post hoc test with significant level set at 5%. M: medium; HP: hydrogen peroxide; TiF${}_{4}$: titanium tetrafluoride.

**Table 1 gels-08-00178-t001:** Treatment groups and composition.

	Group	Abbreviation
35% hydrogen peroxide (control) (Whiteness HP 35% FGM, Joinville, SC, Brazil)	HP (35% hydrogen peroxide, thickener, dyes, glycol, inorganic filler, deionized water)	HP
Experimental group	HP + (TiF4 0.05 g + Natrosol 2.575 g + Chemygel 2.575 g)	HPT0.5
Experimental group	HP + (TiF4 0.1 g + Natrosol 2.55 g + Chemygel 2.55 g)	HPT1
Experimental group	HP + (TiF4 0.2 g + Natrosol 2.5 g + Chemygel 2.5 g)	HPT2
Experimental group	HP + (TiF4 0.3 g + Natrosol 2.45 g + Chemygel 2.45 g)	HPT3
Experimental group	HP + (TiF4 0.4 g + Natrosol 2.4 g + Chemygel 2.4 g)	HPT4

Legend: TiF_4_: titanium tetrafluoride (Sigma-Aldrich Brasil Ltda, Cotia, SP, Brazil, MKCB 9490); HP: 35% hydrogen
peroxide, 040219; Natrosol gel: NatrosolTM Hydroxyethylcellulose, Sigma-Aldrich Brasil Ltda, Cotia, SP, Brazil,
416820; Chemygel: Chemyunion Ltda, Manalapan, NJ, USA, CN006-0518.

**Table 2 gels-08-00178-t002:** Mean and standard deviation (SD) of pH measurement.

Group	Baseline	5 min	10 min	15 min
HP	6.66 (0.3)	6.01 (0.2)	5.83 (0.3)	5.62 (0.2)
HPT0.5	5.43 (0.6)	4.89 (0.5)	4.77 (0.3)	4.82 (0.2)
HPT1	4.35 (0.6)	4.12 (0.6)	4.18 (0.5)	4.29 (0.5)
HPT2	4.16 (0.4)	3.99 (0.1)	4.13 (0.3)	4.20 (0.3)
HPT3	3.66 (0.2)	3.11 (0.7)	3.15 (0.7)	3.19 (0.7)
HPT4	2.50 (0.6)	1.67 (0.4)	1.93 (0.8)	2.10 (0.9)

Legend: HP: hydrogen peroxide; T: titanium tetrafluoride.

## Data Availability

Not applicable.

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
