# Peer review of "A Titanium Tetrafluoride Experimental Gel Combined with Highly Concentrated Hydrogen Peroxide as an Alternative Bleaching Agent: An In Vitro Study"

_gels, 2022, doi:10.3390/gels8030178_

Round 1

Reviewer 1 Report

The aim of the present article was to evaluate the color change, mineral content and morphology of the enamel treated with bleaching agents, and the pH and cytotoxicity of these experimental agents. The effects of bleaching agents tested were investigate by means several modern analytical techniques such as the spectrophotometer, Knoop microhardness test, MTT assay, scanning electron microscopy and energy dispersive spectrometry.

Overall, the manuscript is well-written and well-structured, and it is possible to easily follow the authors’ descriptions and explanations. The organization of the article is good and illustrates the key concepts and conclusions clearly. Throughout the manuscript, I find no fault whatsoever with the methods, data analysis, or conclusions. This study is part of a specific line of research in which the authors are involved, and even if this study is not particularly innovative, in my opinion it represents an interesting contribution in the dental field. I suggest a few minor corrections and comments:

  • The title should be revised and made more concise.
  • Lines 25-28: this period should be separated into two sentences.
  • Line 117: please delete the dash in the title of the subsection "2.5 pH-measurement".
  • Figure 2 (page 6): please added in the second graphic letters to indicate statistical significance.
  • Line 182: Please check the word “PH”
  • Figure 4 (page 7): I suggest increasing the size of the SEM micrographs to make viewing easier.
  • Lines 29,41, 208, 211, 291,292,302, 327, 330, 334: Please check throughout the text that the acronyms “(HP)” are used correctly and not repeated.
  • The Conclusions section should be improved.

According to the journal guidelines, I suggest to decrease self-citation and include references current (mostly within the last 5 years).

In this light, for example replace in Line 34 Ref.12, 13 and 15 with most recent articles. I suggest to read and cite some of these:

  • DOI: 10.5005/jp-journals-10024-2087
  • DOI: 10.7717/peerj.10606
  • DOI: 10.4317/jced.56913

Author Response

RESPONSE TO THE REVIEWERS’ COMMENTS

On behalf of all authors, I acknowledge the comments and contributions made by the reviewers. The detailed responses, together with an account of the changes made to the manuscript, are set out below. A new version of the manuscript (file “Revised-Manuscript”) was uploaded, and all changes in the manuscript are highlighted in red.

Comments from the reviewers:

Reviewer 1

The aim of the present article was to evaluate the color change, mineral content and morphology of the enamel treated with bleaching agents, and the pH and cytotoxicity of these experimental agents. The effects of bleaching agents tested were investigate by means several modern analytical techniques such as the spectrophotometer, Knoop microhardness test, MTT assay, scanning electron microscopy and energy dispersive spectrometry.

Overall, the manuscript is well-written and well-structured, and it is possible to easily follow the authors’ descriptions and explanations. The organization of the article is good and illustrates the key concepts and conclusions clearly. Throughout the manuscript, I find no fault whatsoever with the methods, data analysis, or conclusions. This study is part of a specific line of research in which the authors are involved, and even if this study is not particularly innovative, in my opinion it represents an interesting contribution in the dental field. I suggest a few minor corrections and comments:

RESPONSE: We would like to thank the reviewer for the contributions to our study. All the aspects of concern to the reviewers were addressed, and we hope the revised manuscript achieves the quality standards for publication in the Gels journal.

  • The title should be revised and made more concise.

RESPONSE: The title was revised, and a concise title was provided.

A titanium tetrafluoride experimental gel combined with highly concentrated hydrogen peroxide as an alternative bleaching agent: an in vitro study.

  • Lines 25-28: this period should be separated into two sentences.

RESPONSE: Separation of the sentences was provided.

  • Line 117: please delete the dash in the title of the subsection "2.5 pH-measurement".

RESPONSE: The dash was removed.

  • Figure 2 (page 6): please added in the second graphic letters to indicate statistical significance.

RESPONSE: Letters indicating statistical differences were provided. Specifically in the second graphic of figure 2, there were no statistical differences.

  • Line 182: Please check the word “PH”

RESPONSE: This word was corrected to HP.

  • Figure 4 (page 7): I suggest increasing the size of the SEM micrographs to make viewing easier.

RESPONSE: Figure 4 is composed of all SEM micrographs and EDS graphics, as a representation of enamel surface morphology and enamel surface composition, respectively. Unfortunately, the LaTex system, as recommended by MDPI, has limited the size of all figures used, however, we accept increasing the size of the mentioned figure up to the maximum allowed limit.

  • Lines 29,41, 208, 211, 291,292,302, 327, 330, 334: Please check throughout the text that the acronyms “(HP)” are used correctly and not repeated.

RESPONSE: Thank you for your attention. This acronym was corrected.

  • The Conclusions section should be improved.

RESPONSE: The conclusion section was improved, answering the objective of the study.

  • According to the journal guidelines, I suggest to decrease self-citation and include references current (mostly within the last 5 years).

In this light, for example replace in Line 34 Ref.12, 13 and 15 with most recent articles. I suggest to read and cite some of these:

  • DOI: 10.5005/jp-journals-10024-2087
  • DOI: 10.7717/peerj.10606
  • DOI: 10.4317/jced.56913

RESPONSE: Thank you. We accept the recommendation, and some references were replaced according to the suggestions.

Reviewer 2 Report

The manuscript present original research study with experimental agents containing different concentrations of TiF4 combined with highly concentrated hydrogen peroxide for whitening treatment.

I have a few comments:

  1. What are the limitations of the study?
  2. In my opinion the results were more relevant if the enamel/dentin bovine blocks, after the coloration process in black tea, were stored in artificial saliva, which is an appropriate medium than water.

Author Response

RESPONSE TO THE REVIEWERS’ COMMENTS

On behalf of all authors, I acknowledge the comments and contributions made by the reviewers. The detailed responses, together with an account of the changes made to the manuscript, are set out below. A new version of the manuscript (file “Revised-Manuscript”) was uploaded, and all changes in the manuscript are highlighted in red.

Comments from the editors and reviewers:

Reviewer 2

The manuscript present original research study with experimental agents containing different concentrations of TiF4 combined with highly concentrated hydrogen peroxide for whitening treatment.

I have a few comments:

  • What are the limitations of the study?

RESPONSE: We would like to thank the reviewer for the contributions to our study. All the aspects of concern to the reviewers were addressed, and we hope the revised manuscript achieves the quality standards for publication in the Gels journal.

The limitation of this study was related to cell viability methodology performed in HaCat cells, that represents an evaluation of accidental contact of dental bleaching gel during in-office application technique. Evaluation in odontoblast cells needs to be performed in future studies to determine the protective effect of this experimental titanium tetrafluoride compound-based gel to determine safety and prevent tooth sensibility.

  • In my opinion the results were more relevant if the enamel/dentin bovine blocks, after the coloration process in black tea, were stored in artificial saliva, which is an appropriate medium than water.

RESPONSE: Thank you for your concern. We agree with the reviewer and we will take into consideration for the future in vitro studies. In this manuscript, we performed according to the Sulieman et al. (2003) study and others already published scientific studies, as follows:

Sulieman M, Addy M, Rees JS. Development and evaluation of a method in vitro to study the effectiveness of tooth bleaching. J Dent. 2003 Aug;31(6):415-22. doi: 10.1016/s0300-5712(03)00069-1.

Lins RBE, Rosalen PL, Lazarini JG, Martins LRM, Cavalli V. Assessment of a novel bleaching agent formula containing 35% hydrogen peroxide and titanium tetrafluoride: an in vitro study. Braz Oral Res. 2021 May 31;35:e066. doi: 10.1590/1807-3107bor-2021.vol35.0066.

Palandi SDS, Kury M, Picolo MZD, Coelho CSS, Cavalli V. Effects of activated charcoal powder combined with toothpastes on enamel color change and surface properties. J Esthet Restor Dent. 2020 Dec;32(8):783-790. doi: 10.1111/jerd.12646.
